

# A large Late Miocene cetotheriid (Cetacea, Mysticeti) from the Netherlands clarifies the status of Tranatocetidae

Felix G. Marx[1,2,3,4], Klaas Post[5], Mark Bosselaers[2,6] and Dirk K. Munsterman[7]

[1] Department of Geology, Université de Liège, Liège, Belgium
[2] Directorate of Earth and History of Life, Royal Belgian Institute of Natural Sciences, Brussels, Belgium
[3] Palaeontology, Museums Victoria, Melbourne, Victoria, Australia
[4] School of Biological Sciences, Monash University, Clayton, Victoria, Australia
[5] Natuurhistorisch Museum, Rotterdam, The Netherlands
[6] Zeeland Royal Society of Sciences, Middelburg, The Netherlands
[7] Netherlands Institute of Applied Geoscience TNO - National Geological Survey, Utrecht, The Netherlands

## ABSTRACT

Cetotheriidae are a group of small baleen whales (Mysticeti) that evolved alongside modern rorquals. They once enjoyed a nearly global distribution, but then largely went extinct during the Plio-Pleistocene. After languishing as a wastebasket taxon for more than a century, the concept of Cetotheriidae is now well established. Nevertheless, the clade remains notable for its variability, and its scope remains in flux. In particular, the recent referral of several traditional cetotheriids to a new and seemingly unrelated family, Tranatocetidae, has created major phylogenetic uncertainty. Here, we describe a new species of *Tranatocetus*, the type of Tranatocetidae, from the Late Miocene of the Netherlands. *Tranatocetus maregermanicum* sp. nov. clarifies several of the traits previously ascribed to this genus, and reveals distinctive auditory and mandibular morphologies suggesting cetotheriid affinities. This interpretation is supported by a large phylogenetic analysis, which mingles cetotheriids and tranatocetids within a unified clade. As a result, we suggest that both groups should be reintegrated into the single family Cetotheriidae.

## INTRODUCTION

Cetotheriids are one of three major branches of modern baleen whales, alongside right whales (Balaenidae) and rorquals (Balaenopteridae). The family is first recorded during the Middle Miocene (*Gol'din, 2018*), but its roots likely stretch further back in time (*Marx & Fordyce, 2015*). Late Miocene cetotheriids were speciose and attained a nearly global distribution, with records from the North Atlantic (*Bisconti, 2015*; *Marx, Bosselaers & Louwye, 2016*; *Whitmore Jr & Barnes, 2008*), the Paratethys (*Gol'din & Startsev, 2017*), and both the North (*El Adli, Deméré & Boessenecker, 2014*; *Kellogg, 1929*; *Saita, Komukai & Oishi, 2011*; *Tanaka, Furusawa & Barnes, 2018b*; *Tanaka & Watanabe, 2018*) and eastern South Pacific (*Bouetel & de Muizon, 2006*; *Marx, Lambert & Muizon, 2017*).

Corresponding author
Felix G. Marx,
felix.marx@monash.edu,
felixgmarx@gmail.com

This taxonomic diversity was accompanied by notably disparity, giving rise to at least three distinct morphotypes: (i) Cetotheriinae, a group of small-bodied species closely related to the eponymous *Cetotherium*; cetotheriines were apparently endemic to the Paratethys, and are characterised by a dorsoventrally deep zygomatic process of the squamosal, a notably high (rather than elongate) angular process of the mandible, and a wide, squared anterior border of the tympanic bulla (*Gol'din & Startsev, 2017*); (ii) Herpetocetinae, a second group of small-bodied whales that inhabited the North Atlantic and North Pacific, and came to the fore during the Pliocene; members of this group share a broad exposure of the alisphenoid in the temporal fossa, the presence of a large postparietal foramen, a notably elongate angular process of the mandible, a shelf-like lateral tuberosity of the periotic, and a broad ridge delimiting the posterior border of the facial sulcus on the compound posterior process (*Boessenecker, 2011*; *El Adli, Deméré & Boessenecker, 2014*; *Tanaka, Furusawa & Barnes, 2018b*; *Tanaka & Watanabe, 2018*; *Whitmore Jr & Barnes, 2008*); and (iii) a possibly para- or even polyphyletic group comprising *Herentalia*, *Metopocetus*, and *Piscobalaena*, characterised by the presence of a well-developed posteroventral flange (also present in cetotheriines and herpetocetines, but smaller), a bulbous or indistinct lateral tuberosity of the periotic, a transversely compressed internal acoustic meatus, and a sharp rim surrounding the proximal opening of the facial canal (*Bouetel & de Muizon, 2006*; *Gol'din & Steeman, 2015*; *Marx, Lambert & Muizon, 2017*).

Beyond these morphotypes, the scope of the family remains in doubt. This is partly because of the proposed inclusion of *Cephalotropis* and neobalaenines, which has proved controversial (*Bisconti, 2015*; *El Adli, Deméré & Boessenecker, 2014*; *Fordyce & Marx, 2013*; *Gol'din & Steeman, 2015*; *Marx & Fordyce, 2016*); and partly because of the recent referral of several presumed cetotheriids, such as '*Cetotherium*' *megalophysum* and '*Metopocetus*' *vandelli*, to the new and seemingly unrelated family Tranatocetidae (*Gol'din & Steeman, 2015*).

Tranatocetidae was defined based on *Tranatocetus argillarius*, known only from the Late Miocene clay pit of Gram, Denmark. *Tranatocetus* indeed stands out for its large size, relative to cetotheriids, but its interpretation is severely hampered by the poor preservation of the available material. In particular, crushing and breakage have affected all of the holotype, obliterating details of the otherwise highly diagnostic ear region and necessitating extensive reconstructions of the mandible (*Gol'din & Steeman, 2015*; *Roth, 1978*).

Here, we report a second species of *Tranatocetus*, based on two Late Miocene fossils dredged from the bottom of the Western Scheldt (the Netherlands). The new specimens preserve crucial details that are absent in the type material of *T. argillarius*, and thus offer a perfect opportunity to test the idea that Tranatocetidae and Cetotheriidae form separate clades.

## MATERIAL AND METHODS

### Collection, preparation, body size and phylogenetic analysis

The two specimens described here were trawled from the bottom of the Western Scheldt (the Netherlands) during NMR expeditions 2014-3 and 2015-1 (Fig. 1). Both fossils were

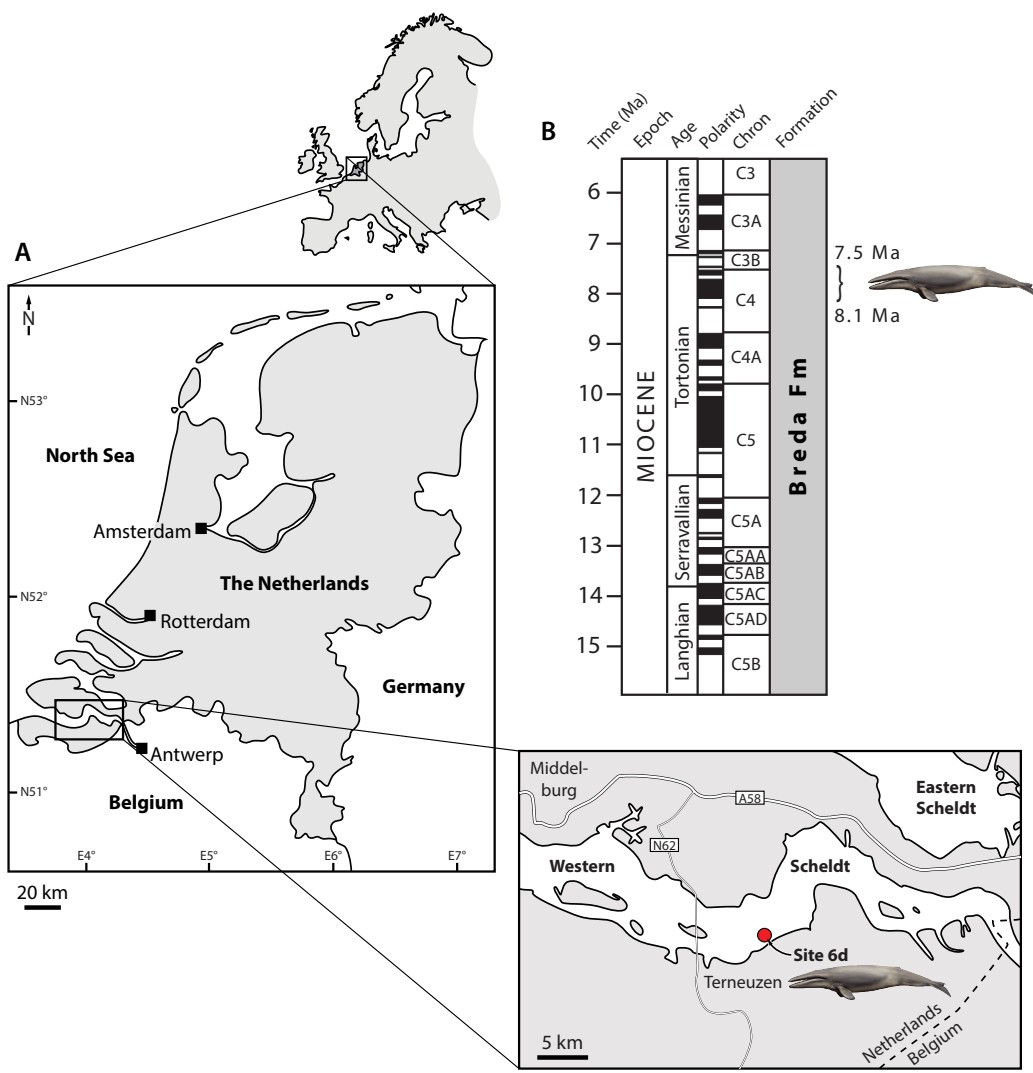

**Figure 1** Type locality (A) and horizon (B) of *Tranatocetus maregermanicum*. Curly bracket in (B) marks the type horizon, as judged from the dinoflagellate fauna associated with the whale fossils. Modified from *Marx, Bosselaers & Louwye (2016)* under a Creative Common Attribution Licence (CC-BY 4.0). Dates in (B) based on *Ogg, Ogg & Gradstein (2016)*. Drawing of cetotheriid by Carl Buell.

embedded in a matrix of hard glauconitic sandstone, and prepared mechanically. For the description, morphological terminology follows *Mead & Fordyce (2009)*, unless indicated. Photographs of the specimens were digitally stacked in Photoshop CS6.

Total body length was estimated based on bizygomatic width, using the stem balaenopteroid equation of *Pyenson & Sponberg (2011)*, and the general mysticete equation of *Lambert et al. (2010)*. To establish evolutionary relationships, we coded our new material, as well as *Tranatocetus argillarius* and the recently described Middle Miocene cetotheriid *Ciuciulea davidi* (*Gol'din & Steeman, 2015*; *Gol'din, 2018*), into a slightly modified version of the phylogenetic matrix of *Fordyce & Marx (2018)*. The analysis was run in MrBayes

3.2.6, on the Cyberinfrastructure for Phylogenetic Research (CIPRES) Science Gateway (*Miller, Pfeiffer & Schwartz, 2010*), using the same settings as in *Fordyce & Marx (2018)*. The full cladistic matrix is available as online supplementary material.

## Age determination

Matrix samples from the more complete specimen (NMR 9991-16680) were prepared at Palynological Laboratory Services (PLS, UK) using standard sample processing procedures, which involve HCl and HF treatment, heavy liquid separation, and sieving over a 15 µm mesh sieve. The organic residue was mounted with glycerine-gelatine on microscopic slides. Two microscopic slides were made: one carrying non-oxidized kerogen, and one on which the organic residue was slightly oxidized with $HNO_3$ to concentrate the palynomorphs and reduce 'Structureless Organic Matter'.

Palynological analysis was carried out at the Geological Survey of the Netherlands (TNO). We counted the first 200 sporomorphs (pollen and spores) and dinoflagellate cysts, and thereafter scanned for any rarer dinocyst species. Major miscellaneous categories (e.g., marine acritarchs, test linings of foraminifers, and the brackish alga *Botryococcus*) were calculated separately. Age interpretations were based on the first and last occurrences of dinoflagellate cysts, using the dinozones of *Munsterman & Brinkhuis (2004)* recalibrated to *Ogg, Ogg & Gradstein (2016)*. Dinoflagellate cyst taxonomy follows the 'Lentin and Williams index' (*Williams, Fensome & MacRae, 2017*).

## Nomenclatural acts

The electronic version of this article in Portable Document Format (PDF) will represent a published work according to the International Commission on Zoological Nomenclature (ICZN), and hence the new names contained in the electronic version are effectively published under that Code from the electronic edition alone. This published work and the nomenclatural acts it contains have been registered in ZooBank, the online registration system for the ICZN. The ZooBank LSIDs (Life Science Identifiers) can be resolved and the associated information viewed through any standard web browser by appending the LSID to the prefix http://zoobank.org/. The LSID for this publication is: urn:lsid:zoobank.org:pub:D39ACC32-687F-4C95-9CD8-A2B17B2DBAFC. The online version of this work is archived and available from the following digital repositories: PeerJ, PubMed Central and CLOCKSS.

## RESULTS

### Systematic palaeontology

Cetacea Brisson, 1762
Neoceti Fordyce and de Muizon, 2001
Mysticeti Gray, 1864
Chaeomysticeti Mitchell, 1989
Cetotheriidae Brandt, 1872
*Tranatocetus Gol'din & Steeman, 2015*

**Type species.** *Tranatocetus argillarius* (*Roth, 1978*)

**Emended diagnosis.** Large cetotheriid sharing with other members of the family the presence of an enlarged compound posterior process of the tympanoperiotic [hereafter, compound posterior process], an enlarged paroccipital concavity extending on to the compound posterior process, elongate, medially convergent ascending processes of the maxillae, a supraoccipital shield whose tip does not extend beyond the apex of the zygomatic process of the squamosal, and a posteriorly projected angular process of the mandible bearing a well-defined fossa for the medial pterygoid muscle. Further shares with all cetotheriids except *Cephalotropis* and *Joumocetus* the posterior telescoping of the ascending process of the maxilla up to, or beyond, the anterior border of the parietal. Differs from all described cetotheriids in its larger size, in having a flattened platform located inside the posterodorsal corner of the mandibular fossa, and in having a mandibular condyle that does not markedly rise above the level of the mandibular neck. Further differs from all cetotheriids except *Herentalia* in having a notably elongate anterior process of the periotic; from all cetotheriids except *Metopocetus* in having a reduced lateral tuberosity; from all cetotheriids except *Herentalia*, *Metopocetus* and *Piscobalaena* in having a sharp cranial rim surrounding the proximal opening of the facial canal; from *Brandtocetus*, *Cetotherium*, *Kurdalagonus* and *Mithridatocetus* in having a larger posteroventral flange of the compound posterior process that completely floors the facial sulcus, a tympanic bulla that is less squared and narrower anteriorly than posteriorly (in ventral view), a more elongate ascending process of the maxilla, and a more gracile zygomatic process of the squamosal; from *Cephalotropis*, *Ciuciulea* and *Joumocetus* in lacking a long exposure of the parietal on the vertex; from *Cephalotropis* in having a proportionally larger bulla, and a better-developed posteroventral flange of the compound posterior process; from *Herpetocetus* and *Piscobalaena* in having a squamosal cleft; from *Piscobalaena* in having a sharper vomerine crest; from *Herpetocetus* in lacking a postparietal foramen; from '*Cetotherium*' *megalophysum* in having a distally larger compound posterior process with a better-developed posteroventral flange, and a more concave supraoccipital shield; from *Metopocetus* in having a more elongate ascending process of the maxilla, a narrower posterior portion of the nasal, and a smaller tympanohyal; and from *Herentalia* in having a more pointed apex of the supraoccipital shield and in lacking a well-developed external occipital crest.

*Tranatocetus maregermanicum*, sp. nov.
Figs. 2–8

**LSID.** urn:lsid:zoobank.org:act:499F1C5C-3C3F-48A9-AD97-AF19F99DE886

**Holotype.** NMR9991-16680, partial cranium comprising the braincase and ear bones, posterior portions of both mandibles, a fragmentary atlas, and two thoracic vertebrae.

**Paratype.** NMR9991-16681, basicranium, right periotic, atlas, and seventh cervical vertebra.

**Locality and horizon.** Both fossils were recovered from the Breda Formation, exposed at site 6d (N51°21′56.9″, E3°54′25.1″) of *Post & Reumer (2016)*. Assemblages from the

associated matrix are relatively rich in marine dinoflagellate cysts, but include few (ca 10%) sporomorphs (Table S1). The latter mostly consist of bisaccate pollen (71%), which are relatively buoyant and, along with the abundance of dinocysts, indicate a distal position from the coast. The most abundant dinocyst genus is *Spiniferites* (39% of the total dinocyst sum), which preferentially occurs in open marine conditions. However, the temperate–tropical, inner neritic *Barssidinium graminosum* and the coastal *Lingulodinium machaerophorum* are also well-represented (24% and 6%, respectively), suggesting overall neritic conditions.

*Enneadocysta pectiniformis* and *Glaphyrocysta* spp. are reworked from the Oligocene or older intervals. Among the age-diagnostic taxa, *Hystrichosphaeropsis obscura* last occurs in Zone SNSM14 (ca. 7.5 Ma), and defines both the DN9 Zone of *de Verteuil & Norris (1996)* and the *Hystrichosphaeropsis obscura* Zone of *Dybkjær & Piasecki (2010)*. The presence of *Selenopemphix armageddonensis* confirms a date no older than late Tortonian. The first occurrence of this species has variously been placed at either 7.5 Ma (Zone DN10; *De Verteuil & Norris, 1996*; *Dybkjær & Piasecki, 2010*), or at 9 Ma in equatorial areas (*Williams et al., 2004*). In Belgium, it has been recorded from the Kasterlee Formation (*Louwye & de Schepper, 2010*), whereas in the Netherlands it extends to the top of Zone SNSM13, which dates to approximately 8.1 Ma (well Groote Heide). Together, these observations constrain the current assemblage to Zone SNSM14, Late Miocene, ca. 8.1–7.5 Ma (*Munsterman & Brinkhuis, 2004*, recalibrated to *Ogg, Ogg & Gradstein, 2016*).

**Diagnosis.** Shares with *Tranatocetus argillarius* its large overall size, slender ascending processes of the maxillae that are situated centrally on a triangular platform formed by the frontals, a narrow but continuous exposure of the nasals between the ascending processes of the maxillae, the lack of an external occipital crest, a bulbous exoccipital, an elongate anterior process of the periotic lacking a lateral tuberosity, a large paroccipital concavity, a large mandibular fossa housing a flattened platform in its posterodorsal corner, a deep subcondylar furrow, and a mandibular condyle that does not markedly rise above the level of the mandibular neck. Differs from *T. argillarius* in having a more anterolaterally directed base of the supraorbital process of the frontal, a more robust zygomatic process of the squamosal, relatively larger occipital condyles projecting posteriorly beyond the posterior apex of the nuchal crest, a vomerine crest extending posteriorly far beyond the level of the subtemporal crest, and a sharper, dorsally convex ventral border of the mandibular foramen.

**Etymology.** After the Latin name of the North Sea, *Mare Germanicum*, which *Tranatocetus* once inhabited.

## DESCRIPTION

**Overview.** The posterior portion of the skull of NMR9991-16680 is nearly complete, except for the zygomatic processes of the squamosals (Fig. 2). The surface of the vertex is somewhat eroded, and the rostral bones have become detached and have slid forwards along their respective sutures. The rostrum and the supraorbital processes of the frontals are mostly missing. Ventrally, the posterior halves of both mandibles are nearly *in situ*. Anteriorly,

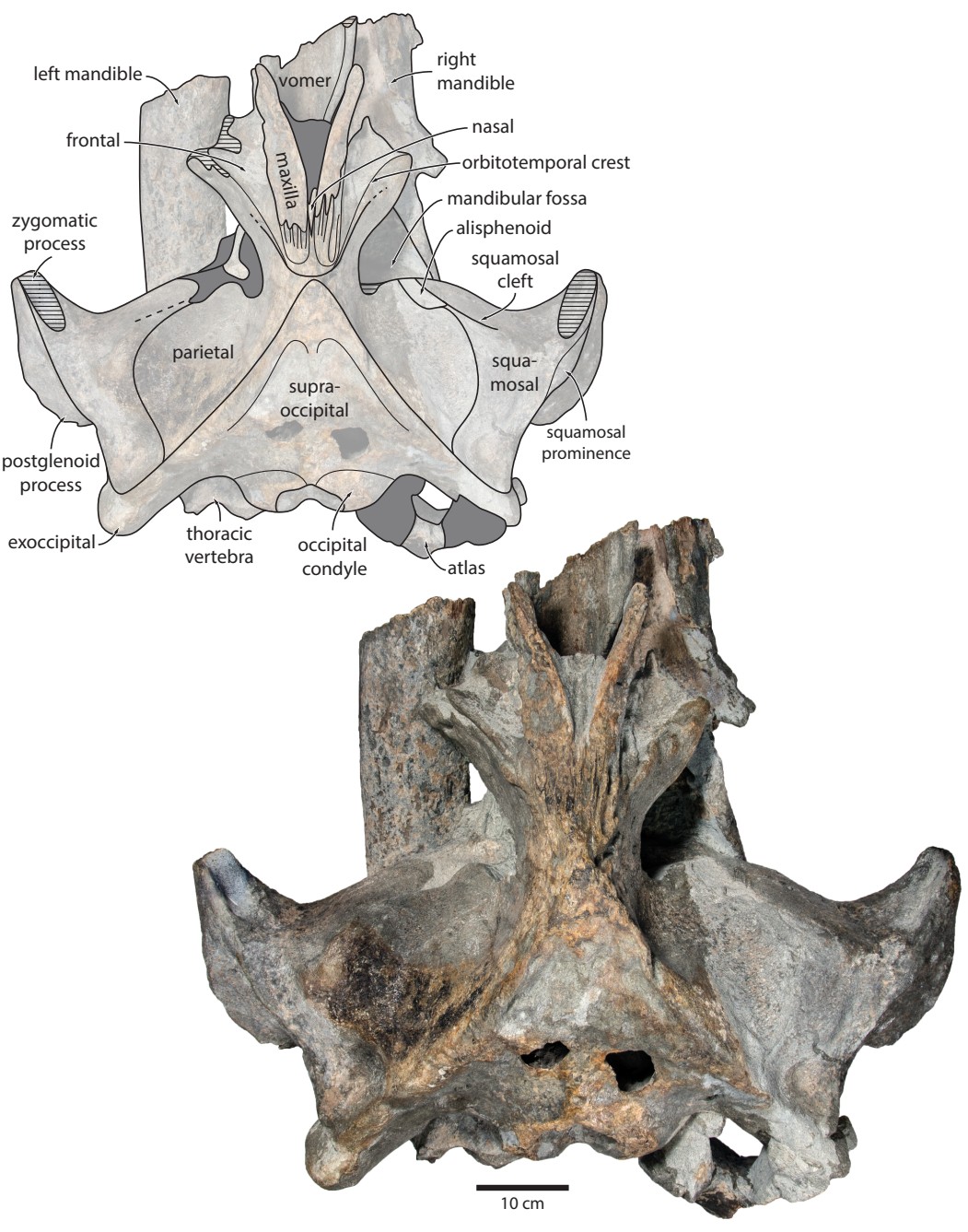

**Figure 2** Holotype cranium of *Tranatocetus maregermanicum* (NMR9991-16680) in dorsal view. Photograph by Felix G. Marx.

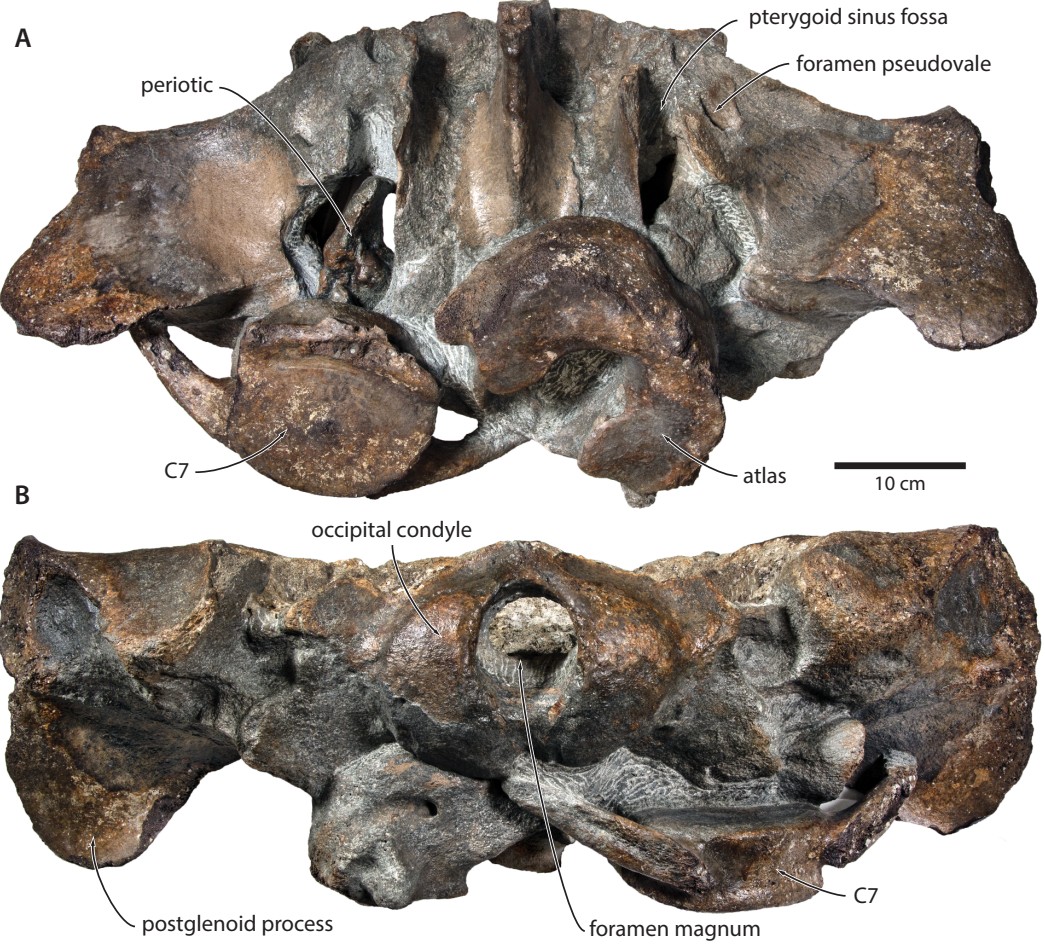

**Figure 3** **Paratype cranium of *Tranatocetus maregermanicum* (NMR9991-16681) in (A) ventral and (B) posterior view.** Photographs by Felix G. Marx.

the mandibles and the vomer are truncated by a continuous oblique fracture, suggesting that the specimen was broken, and its anterior half lost, during dredging. Posteriorly, the atlas and a single thoracic vertebra still adhere to the skull. A second thoracic vertebra was removed during preparation. NMR9991-16681 consists of a well-preserved basicranium, with the right periotic *in situ* but partially obscured (Fig. 3). See Table 1 for measurements of both specimens.

**Maxilla, premaxilla and nasal.** In dorsal view, the ascending process of the maxilla is narrow, parallel-sided, and elongate (Fig. 1). Posteriorly, the ascending processes converge, but appear to remain separated by a thin sliver of nasal. The premaxilla is missing but, judging from the lack of space between the nasal and the maxilla, did not contact the frontal on the vertex. Except for a narrowly exposed section between the posterior maxillae, the nasals are lost and/ or obscured by sediment. In lateral view, the maxilla gently descends from the vertex, suggesting an overall concave facial profile (Figs. 4, 5). The posteriormost portion of the maxilla extends posteriorly beyond the level of the coronal suture.

**Table 1  Measurements of *Tranatocetus maregermanicum* (in mm).**

**NMR NMR9991-16680 (holotype)**

| | |
|---|---|
| Bizygomatic width | 860 |
| Width across exoccipitals | 590 |
| Bicondylar width | 190 |
| Width of foramen magnum | 81 |
| Height of foramen magnum | 54 |
| Width across parietals at intertemporal constriction | 130 |
| Width of ascending process of maxilla (left) | 48 |
| Length of compound posterior process | 107 |
| Width of distal exposure of compound posterior process | 75 |
| Height of distal exposure of compound posterior process | 40 |
| Diameter of external acoustic meatus | 34 |
| Length of tympanic bulla | 99 |
| Width of tympanic bulla at sigmoid process | 82 |
| Width of atlas | 224 |
| Length of atlas | 75 |

**NMR9991-16681 (paratype)**

| | |
|---|---|
| Bicondylar width | 190[a] |
| Width of foramen magnum | 60 |
| Height of foramen magnum | 70 |
| Length of periotic (anterior process + body) | 113[a] |
| Width of periotic | 54 |
| Height of periotic | 43 |
| Length of pars cochlearis | 40 |
| Width of pars cochlearis | 24 |
| Width of atlas | 244[a] |
| Length of atlas | 81 |
| Height of atlas | 180 |

**Notes.**
[a] estimated.

**Vomer.** In anterior view, the fractured vomer is V-shaped in cross section and floors the mesorostral groove (Fig. 5). In ventral view, the ventral portion of the vomer gradually flares at the level of the temporal fossa to form a lozenge-shaped platform (Fig. 6). Posterior to this platform, the tall but rounded vomerine crest ascends towards the braincase, and eventually merges with the plate-like posteriormost portion of the vomer that separates the pharyngeal crests.

**Frontal.** In dorsal view, the frontal is nearly excluded from the vertex, and forms a V-shaped suture with the parietal (Fig. 2). There is no obvious narial process. A well-defined orbitotemporal crest originates on the vertex, and from there runs close and nearly parallel to the posterior border of the supraorbital process. Medially, this crest is separated from the ascending process of the maxilla by an anteriorly widening triangular basin. In anterior

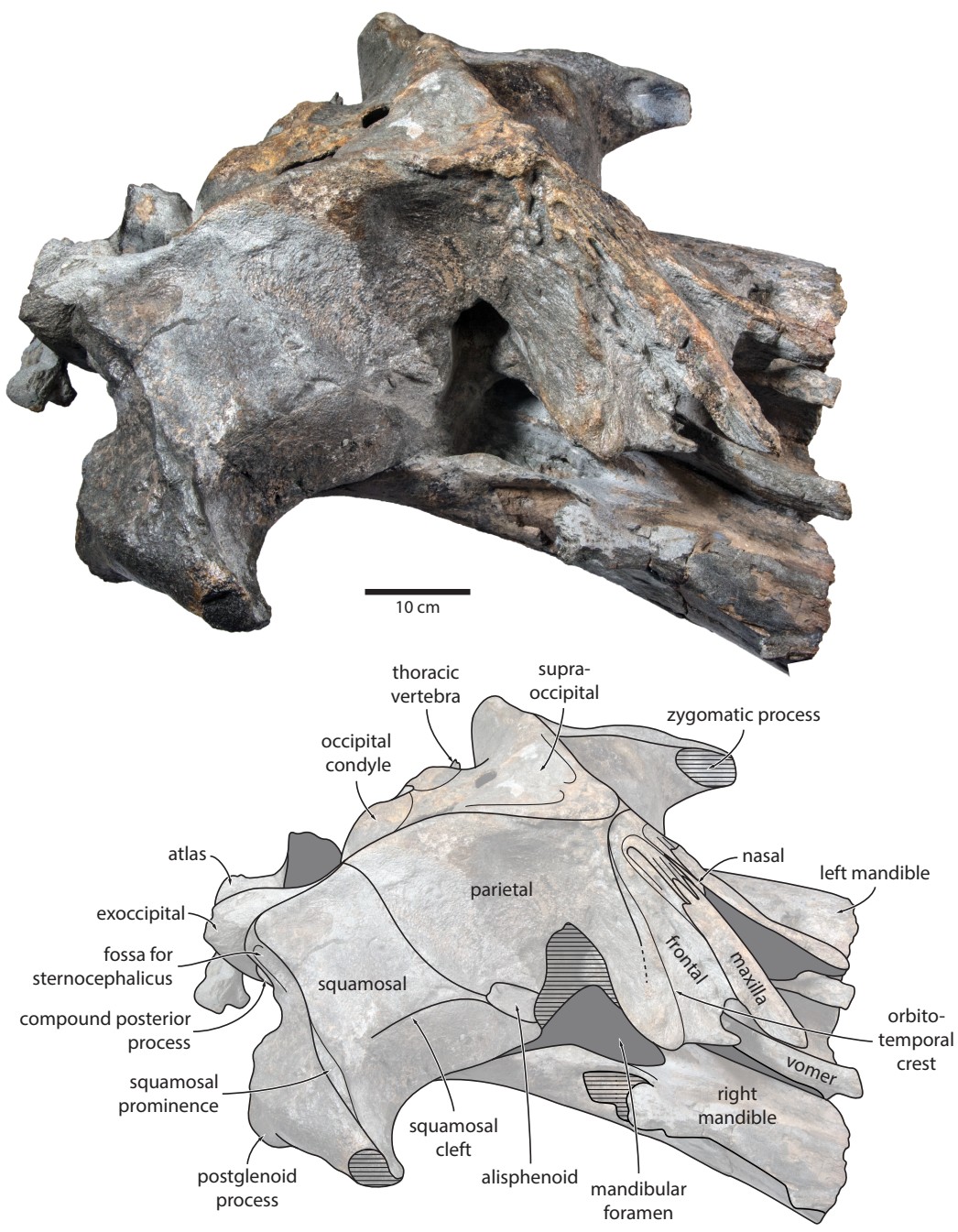

**Figure 4** **Holotype cranium of *Tranatocetus maregermanicum* (NMR9991-16680) in oblique right anterolateral view.** Photograph by Felix G. Marx.

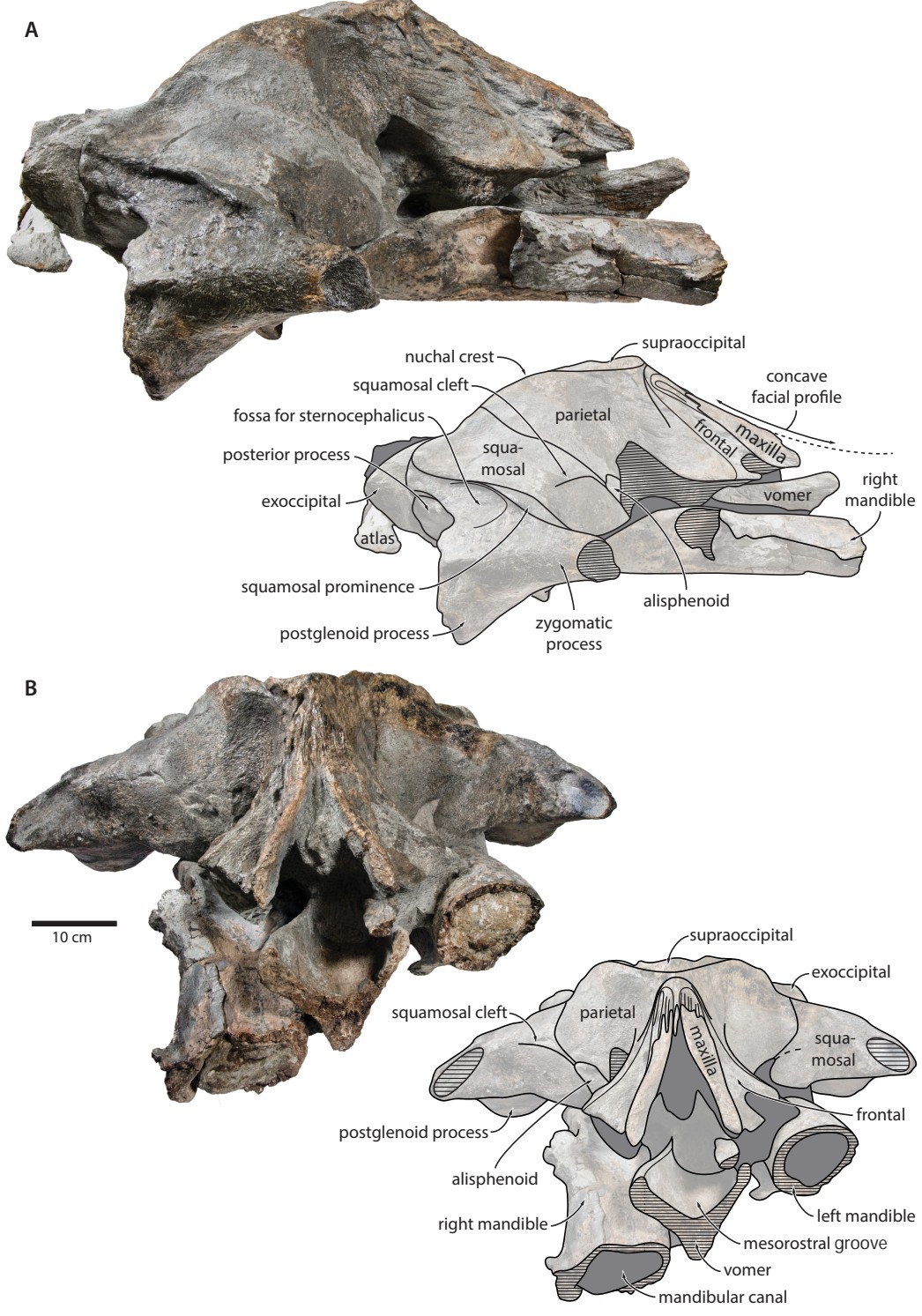

**Figure 5** **Holotype cranium of *Tranatocetus maregermanicum* (NMR9991-16680) in (A) lateral and (B) anterior view.** Photographs by Felix G. Marx.

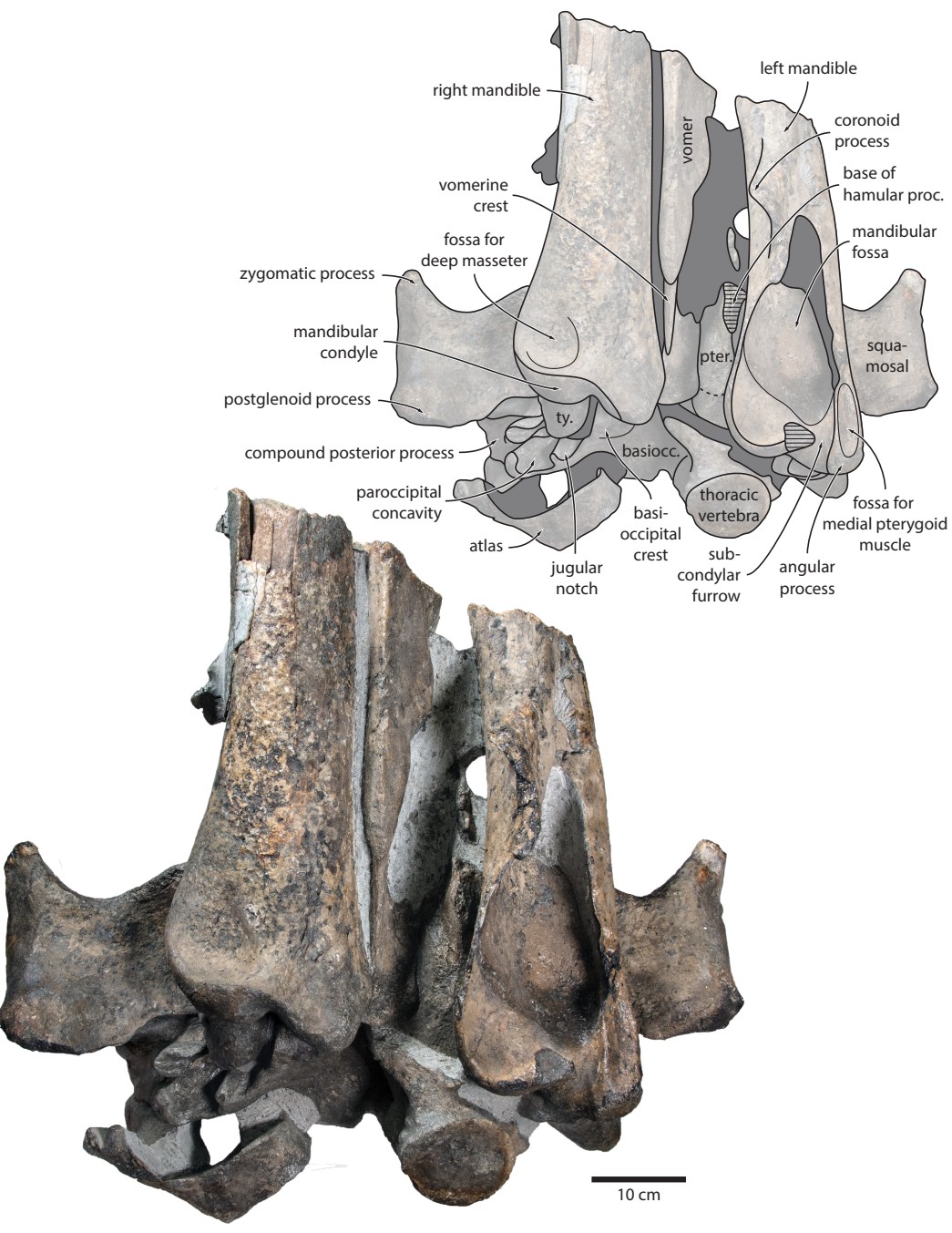

**Figure 6** **Holotype cranium of *Tranatocetus maregermanicum* (NMR9991-16680) in ventral view.** basiocc., basioccipital; pter., pterygoid; hamular proc., hamular process; ty., tympanic bulla. Photograph by Felix G. Marx.

view, the base of the supraorbital process is descending gradually from the vertex, with its dorsal border being notably concave (Fig. 5).

**Parietal.** In dorsal view, the parietal appears to be virtually excluded from the vertex, but it (or the interparietal) still likely contributes to the apex of the supraoccipital shield (Fig. 2). Anteriorly, a nearly vertical, triangular 'wing' of the parietal overrides the posteromedial portion of the frontal and underlaps the orbitotemporal crest (Fig. 4). Unlike in *Herentalia*, *Metopocetus* and *Piscobalaena*, the anterodorsal border of the parietal does not flare laterally, and hence does not 'buttress' the vertex. The parieto-squamosal suture is sigmoidal: after descending from the nuchal crest, it turns first anterolaterally and then anteromedially, before terminating at the presumed position of the alisphenoid. Along the suture, the parietal and squamosal slightly bulge into the temporal fossa. There is no tubercle at the junction of the parieto-squamosal suture with the nuchal crest, and no postparietal foramen (Fig. 4).

**Squamosal.** In dorsal view, the temporal fossa is wider than long (Fig. 2). The squamosal fossa is elongate, and approximately as long as the temporal fossa is wide. The base of the zygomatic process is oriented somewhat anterolaterally, and bears a gently rounded supramastoid crest. On the right, there is a low but clearly defined squamosal prominence. The posterior apex of the nuchal crest is located anterior to the level of the occipital condyles. A squamosal cleft is present and extends laterally from the presumed location of the alisphenoid. There is no squamosal crease.

In ventral view, the base of the postglenoid process is oriented transversely, with no obvious twisting (Figs. 3 and 6). The glenoid fossa is smooth and not visibly offset from the surrounding bone. The fossa for the sigmoid process of the tympanic bulla is indistinct. Medially, the postglenoid is confluent with a low anterior meatal crest delimiting the proximal portion of the external acoustic meatus. The posterior meatal crest descends along the anterior face of the compound posterior process, and extends laterally on to the posterior face of the postglenoid process. The falciform process is robust, hook-shaped and separated from the spinous process by an approximately rectangular fenestra, similar to *Metopocetus*. The foramen pseudovale is almost entirely surrounded by the squamosal, except for a narrow sliver of pterygoid that contributes to its anterior border.

In lateral view, the zygomatic process is robust and approximately as tall as it is wide transversely (Fig. 5). The postglenoid process is triangular, with a vertical posterior face and a posteroventrally oriented anterior border. Anterodorsal to the compound posterior process, and immediately below the supramastoid crest, there is a large fossa for the sternocephalicus that partially excavates the supramastoid crest. In posterior view, the postglenoid process is approximately parabolic in outline, and descends ventrally well below the level of the exoccipital (Fig. 3).

**Alisphenoid.** The alisphenoid remains obscured by matrix, but a depression in the temporal wall, between the parietal dorsally and the squamosal ventrally, marks its likely position, and suggests that it may have contributed to the orbital fissure (Fig. 4).

**Supraoccipital, exoccipital, basioccipital.** In dorsal view, the supraoccipital is sharply triangular, and extends anteriorly beyond the level of the subtemporal crest (Fig. 2). The nuchal crest is oriented dorsolaterally but does not overhang the temporal fossa.

Between the nuchal crests, the supraoccipital shield is initially flattened near its apex, but then becomes moderately concave transversely. Its surface is largely eroded, but seems to have lacked a well-developed external occipital crest. The exoccipital is robust, thickened anteroposteriorly, and extends posteriorly well beyond the level of the occipital condyles. The condyles themselves are robust and lack a distinct neck.

In posterior view, the foramen magnum is rounded, and approximately half as high as the occipital condyles (Fig. 3). The paroccipital process is concave ventrally, slightly offset from the more lateral portion of the exoccipital by a blunt notch, and descends to approximately the same level as the basioccipital crest.

In ventral view, the basioccipital crest is robust and approximately triangular (Figs. 6 and 7). The jugular notch is relatively wide and oriented ventrolaterally. The paroccipital concavity is enormous, and medially excavates the bony wall separating it from the jugular notch (Fig. 7). The posterior border of the paroccipital concavity is thin, but then markedly thickens laterally and protrudes outwards. Anteriorly, the roof of the paroccipital concavity is uneven, with a noticeably raised centre; the anterior border of the concavity closely approximates the compound posterior process.

**Periotic.** In anterior view, the anterior process is curved dorsoventrally, with the medial face being somewhat concave and the outer surface convex. In medial view, the anterior process is approximately squared, but sediment obscures its precise outline.

In ventral view, the anterior process is robust, elongate, and more than 1.5 times as long as the pars cochlearis (Fig. 7). The anterior process and body of the periotic remain nearly constant in width anteroposteriorly, with no visible hypertrophy. The ventral border of the anterior process bears a sharp keel, which posteriorly terminates in the fused anterior pedicle of the tympanic bulla. The lateral tuberosity and anteroexternal sulcus are indistinct. The pars cochlearis is globular, and anterodorsally bears a shallow depression. Anteroventral to the pars cochlearis, there is a short, robust ridge for the tensor tympani. The mallear fossa is deep, but poorly defined. Posterior to the mallear fossa, there is a bulbous, low squamosal flange.

The compound posterior process is enlarged, plug-like, and clearly exposed on the outer skull wall (Fig. 7). Its distal surface is flattened, and clearly offset from both the ventral face of the process and the facial sulcus. The facial sulcus is floored by a large posteroventral flange (sensu *Marx, Bosselaers & Louwye, 2016*), which widens and thickens externally, and forms the anterior extension of the paroccipital concavity. Medially, the thickened outer portion of the posteroventral flange is delimited by a notch, presumably marking the position of the posteroventral sulcus (sensu *Marx, Lambert & Muizon, 2017*). Anteriorly, the expanded paroccipital concavity is delimited by a robust, posteroventrally curving anteroventral flange.

**Tympanic bulla.** The right bulla is *in situ* and partially covered by the mandible (Figs. 6 and 7). In ventral view, its anterior portion is somewhat narrower than its posterior half. The ventral surface of the bulla is convex transversely throughout. In lateral view, the lateral furrow is approximately vertical. The sigmoid process is straight and lacks a distinct ventral border. The conical process is convex dorsally, and located entirely posterior to the sigmoid process. In posterior view, there are well-developed inner and

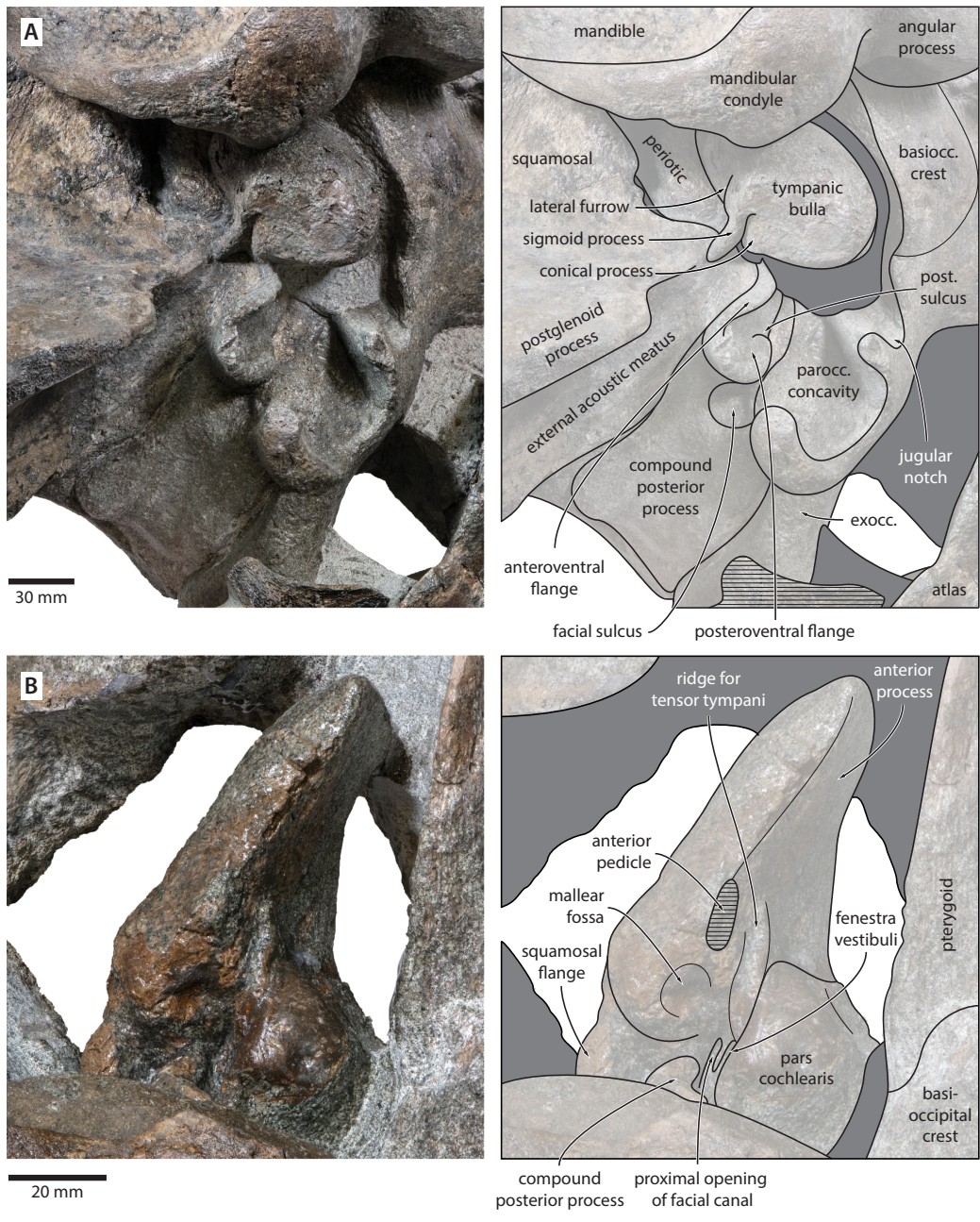

**Figure 7 Auditory anatomy of _Tranatocetus maregermanicum._** (A) Auditory region of the holotype cranium (NMR9991-16680) in oblique right posterolateral view. (B) Periotic of the paratype (NMR9991-16681) in ventral view. exocc., exoccipital; post. sulcus, posteroventral sulcus. Photographs by Felix G. Marx.

outer posterior prominences, separated from each other by a shallow median furrow. As in other crown mysticetes, the bulla has rotated medially, so that the main ridge faces medially towards the basioccipital crest. Other morphological details remain obscured by the mandible.

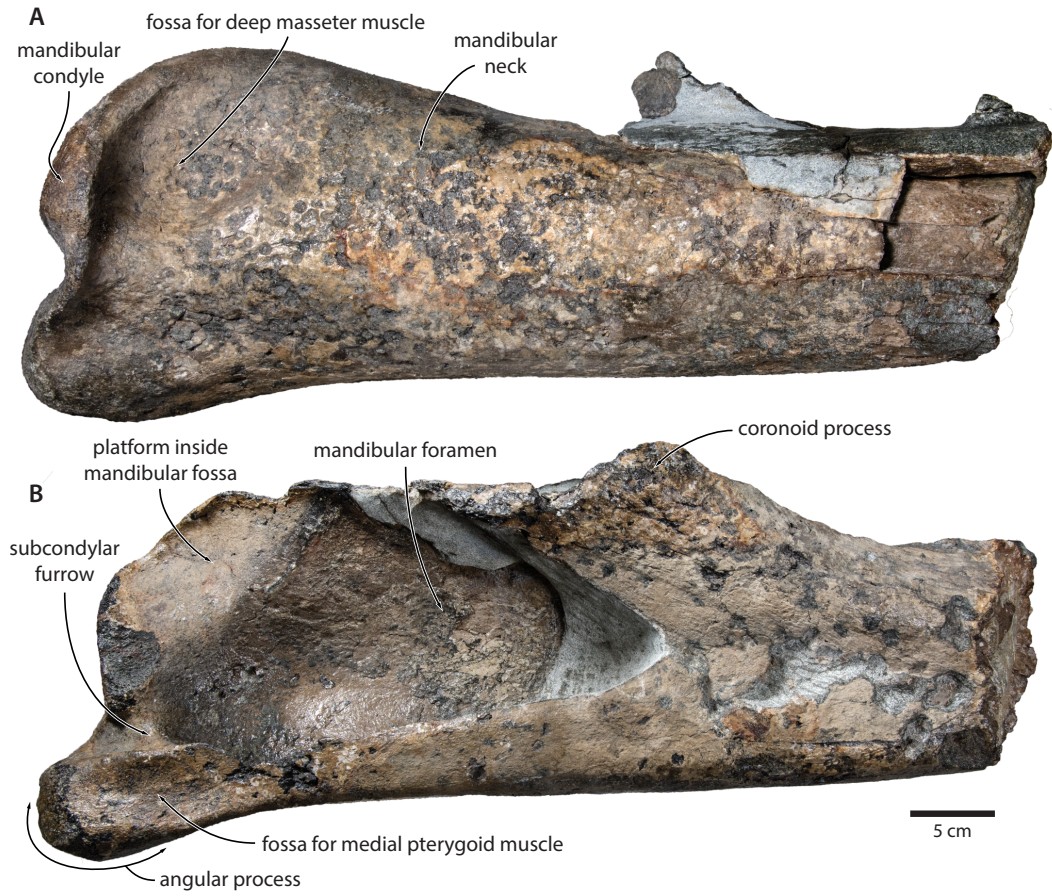

**Figure 8   Holotype mandible of *Tranatocetus maregermanicum* (NMR9991-16680).** (A) Right mandible in lateral view; (B) left mandible in medial view. Photographs by Felix G. Marx.

**Mandible.** In medial view, the coronoid process is low and broadly triangular (Fig. 8). The mandibular foramen is tall dorsoventrally, thus forming a mandibular fossa; it is framed by a robust ventral border, and anteriorly reaches the level of the coronoid process. The angular process is massive, projects posteriorly beyond the level of the mandibular condyle, and bears an elongate fossa for the attachment of the medial pterygoid muscle. The subcondylar furrow is deep and well defined. The medial surface of the condyle is somewhat excavated, and forms a platform occupying the posterodorsal corner of the mandibular fossa.

In lateral view, the condyle is approximately aligned with mandibular neck. Its articular surface points largely posteriorly, but is confluent with a thickened, posterodorsally oriented ridge (Fig. 8). Anterior to the condyle, the lateral surface of the mandible is excavated by a large fossa for the attachment of the deep masseter muscle. The subcondylar furrow is visible as a distinct notch in the posterior profile of the ramus, but does not extend on to its lateral surface. Lateral to the subcondylar furrow, the condyle and angular process

are connected by a sharp crest. The angular process gently descends below the level of the ventral border of the mandibular neck.

In dorsal view, the mandibular body is flattened medially, but dorsoventrally convex laterally. The tip of the coronoid process is strongly bent outwards, suggesting that, in life, the adducted mandible was rotated towards the lateral edge of the rostrum. Posterior to the coronoid process, the mandibular foramen is overhung by a moderately developed postcoronoid elevation. The mandibular neck is straight, rather than recurved as in balaenopterids.

**Postcrania.** In ventral view, the atlas is notably robust. The hypapophysis is reduced to a small protuberance of roughened bone that, judging from its surface texture, may have been weakly fused to the axis. In posterior view, the remainder of the articular surface for the axis, including the well-defined fossa for the odontoid process, is smooth (Fig. 3).

In posterior view, the body of the seventh cervical vertebra is approximately squared. The upper transverse process is relatively slender and oriented anteroventrally. There is no lower transverse process. The posteroventral border of the body is roughed, spongy and somewhat broken, suggesting partial and—presumably—pathological fusion of C7 and T1 (Fig. 3).

In posterior view, the body of the more anterior thoracic vertebra (presumably T3 or T4) is approximately oval (Fig. 6). The transverse process is short and oriented somewhat anteriorly. The body of the more posterior thoracic is far longer anteroposteriorly, and approximately heart-shaped in anterior view. There is no ventral keel. A small anterolateral protuberance approximately halfway up the height of the body presumably represents a semi-facet for the associated rib.

## DISCUSSION AND CONCLUSIONS

### Identification of the new material as *Tranatocetus*

*Tranatocetus maregermanicum* closely resembles *T. argillarius* in its (i) overall size, (ii) slender ascending process of the maxilla, (iii) lack of an external occipital crest, (iv) elongate anterior process of the periotic, (v) lack of a lateral tuberosity on the periotic, (vi) low mandibular condyle, (vii) large mandibular fossa, (viii) presence of a posterodorsal platform inside the mandibular fossa, and (ix) deep subcondylar furrow (Fig. 9).

Nevertheless, the original description of *T. argillarius* also lists several features that appear to differentiate the two species. Of these, the most obvious include a smaller angular process of the mandible which seemingly does not extend beyond the level of the mandibular condyle (*Gol'din & Steeman, 2015*, p. 15–12); an even larger mandibular fossa (*Gol'din & Steeman, 2015*, p. 7); a smaller distal exposure of the compound posterior process (*Gol'din & Steeman, 2015*, p. 4); and a somewhat rounded, rather than straight, lateral border of the supraoccipital (*Gol'din & Steeman, 2015*, Fig. 1).

Our re-examination of *T. argillarius* suggests that all of these differences can be explained by the poor state of preservation of the holotype (GMUC VP2319). Thus, the angular process of the latter is not preserved, and has been entirely reconstructed in resin, leaving its shape and size in doubt (*Roth, 1978*) (Fig. 10). A morphology similar

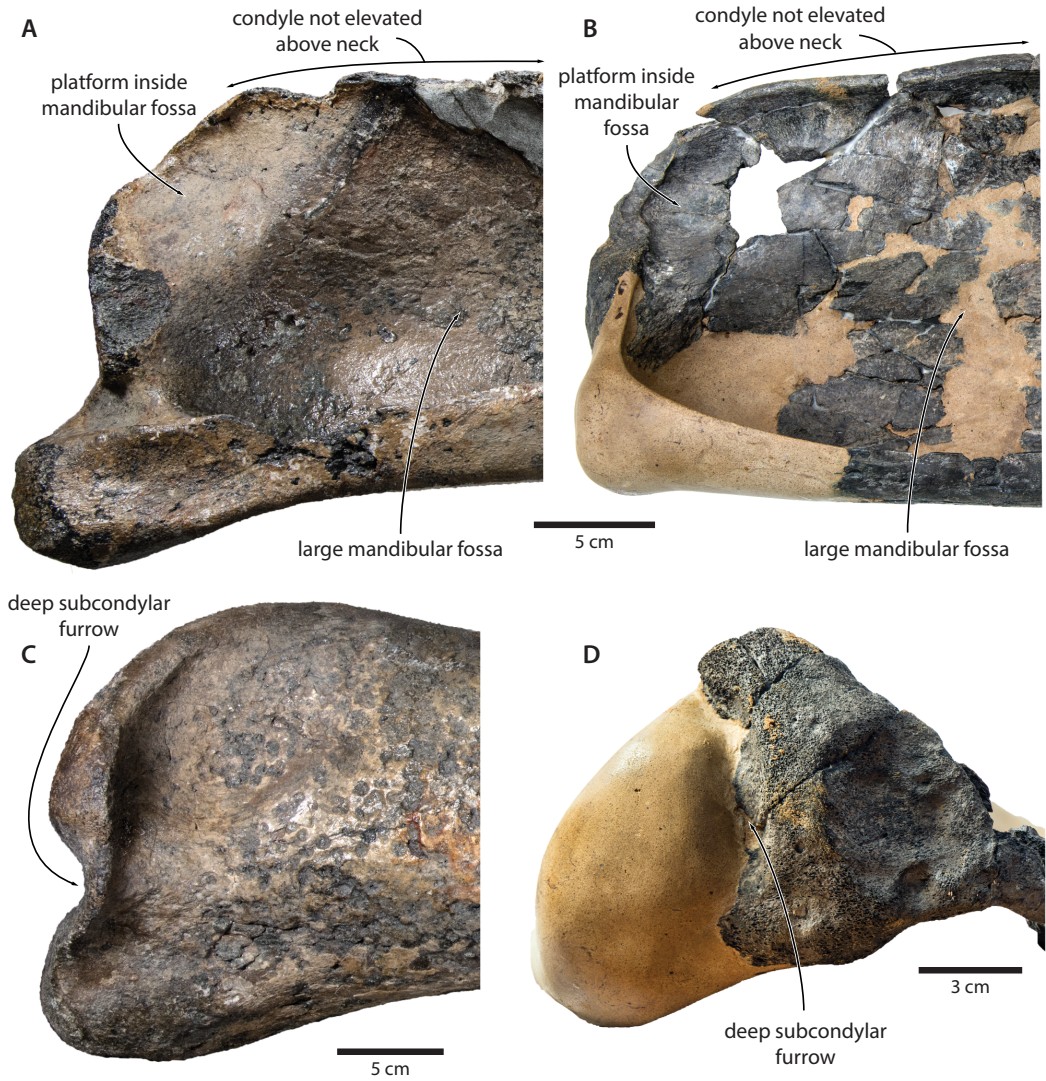

**Figure 9** **Comparison of the mandibular ramus of (A, C)** *Tranatocetus maregermanicum* **(NMR9991-16680, holotype) and (B, D)** *Tranatocetus argillarius* **(GMUC VP2319, holotype).** (A, B) medial, (C) lateral and (D) posterior view. Photographs by Felix G. Marx.

to that of *T. maregermanicum* therefore cannot be excluded, and perhaps might even be indicted by the similarly pronounced subcondylar furrow in both species.

The size and shape of the mandibular fossa are similarly problematic, as its ventral portion in *T. argillarius* has broken off, and no longer makes direct contact with the remainder of the mandible. During reconstruction, it was fixed into its inferred position with resin, but likely somewhat out of place, and at the wrong angle (Fig. 10). In anterior view, the outer wall of the mandibular fossa curves inwards and becomes markedly thinner towards its ventral border. By contrast, the now juxtaposed ventral portion of the mandible (including the ventral border of the mandibular fossa) is notably less concave and relatively thick, implying that it is not in its original position. Extrapolating the curvature of the outer

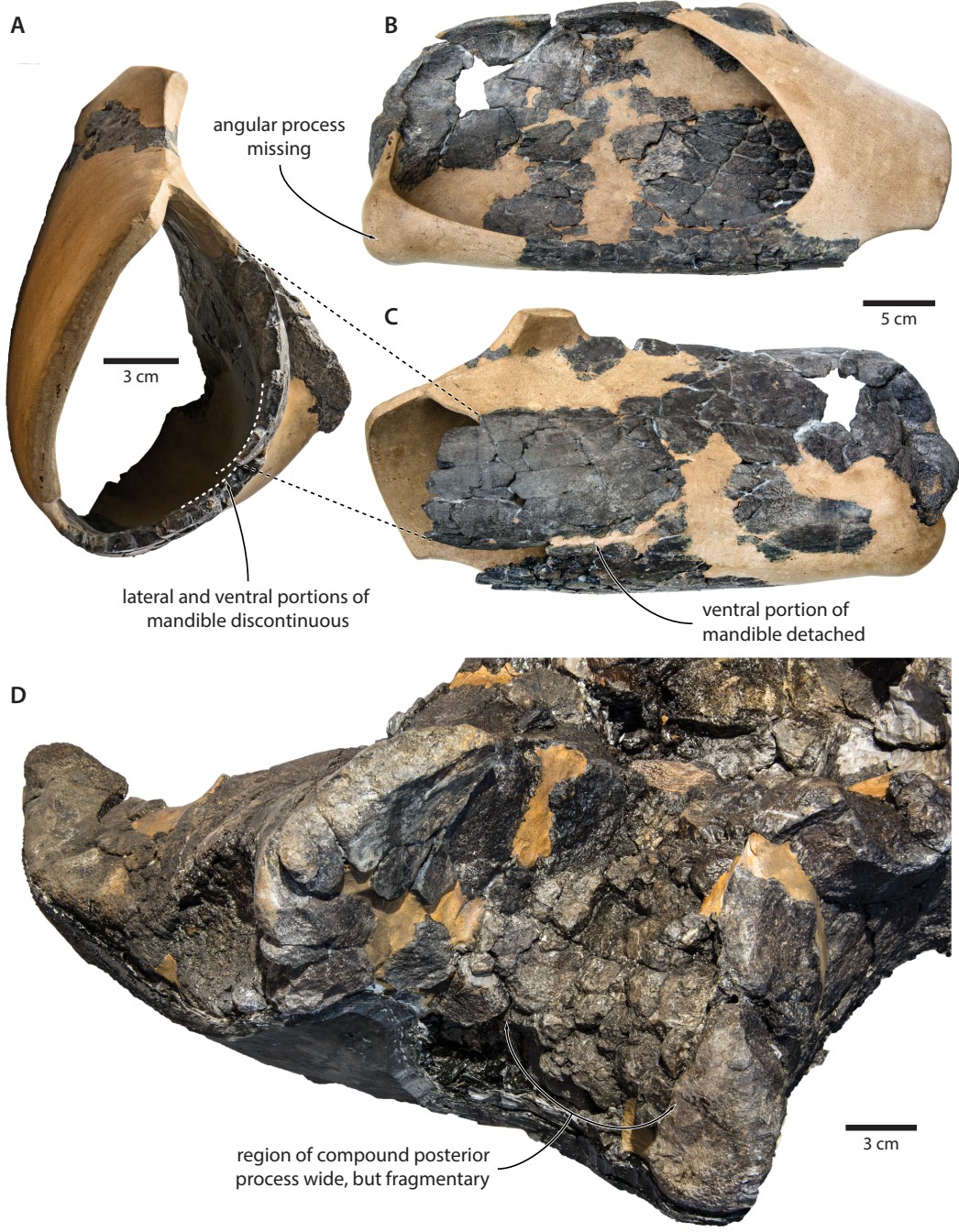

A

B

angular process
missing

C

3 cm

5 cm

lateral and ventral portions of
mandible discontinuous

ventral portion of
mandible detached

D

region of compound posterior
process wide, but fragmentary

3 cm

**Figure 10** **Damage to the mandible and auditory region of *Tranatocetus argillarius* (GMUC VP2319, holotype).** Mandible in (A) anterior, (B) medial and (C) lateral view. (D) Auditory region in oblique right posterolateral view. Photographs by Felix G. Marx.

wall suggests that the cross section of the ramus, and therefore also the mandibular fossa, would originally have been smaller, and thus more similar to that of *T. maregermanicum*.

The compound posterior process of *T. argillarius* is extremely poorly preserved on both sides of the skull (Fig. 10). Despite repeated attempts by two of the authors (FGM and KP), we were unable to trace its outline, thus invalidating it as a diagnostic feature. We note, however, that the space between the external acoustic meatus and the exoccipital is large, and thus consistent with a broadly exposed compound posterior process as seen in *T. maregermanicum*. A large compound posterior process would furthermore match the sizeable paroccipital concavity.

Finally, the supraoccipital is highly fragmentary, which makes its shape difficult to assess. The tip is fixed in place by a large amount of resin, giving rise to an artificially rounded left lateral outline in dorsal view. The right lateral border is comparatively straight, although the supraoccipital shield as a whole still seems somewhat broader and blunter than in *T. maregermanicum*. On the whole, the detailed morphology of this feature likely differs between *T. argillarius* and *T. maregermanicum*, but not as much as the reconstructed skull of the former might suggest.

Based on these observations, we conclude that the features distinguishing the two species of *Tranatocetus* are relatively mild, with the most pronounced differences being attributable to artefacts of preservation. In keeping with the results of our phylogenetic analysis (see below), we therefore reaffirm their placement in the same genus.

## Phylogeny and status of Tranatocetidae

For more than a decade, there has been broad agreement on the basic concept of Cetotheriidae (*Bouetel & de Muizon, 2006*; *El Adli, Deméré & Boessenecker, 2014*; *Gol'din & Startsev, 2017*; *Marx & Fordyce, 2015*; *Steeman, 2007*; *Whitmore Jr & Barnes, 2008*). Nevertheless, the scope of the family has been thrown in doubt by the inclusion of the pygmy right whale, *Caperea marginata* (*Fordyce & Marx, 2013*; *Marx & Fordyce, 2016*; *Park et al., 2017*), and the proposed grouping of several species usually regarded as cetotheriids into the separate family Tranatocetidae (*Gol'din & Steeman, 2015*). The latter is thought to be more closely related to balaenopterids than to cetotheriids, and comprises the eponymous *Tranatocetus*, as well as *Mesocetus longirostris*, *Mixocetus elysius*, '*Aulocetus latus*', '*Cetotherium*' *megalophysum*, and '*Metopocetus*' *vandelli* (*Gol'din & Steeman, 2015*). Of these, the last three frequently cluster in phylogenetic analyses, and may represent the same genus or even species (*El Adli, Deméré & Boessenecker, 2014*; *Gol'din, 2018*; *Marx, Bosselaers & Louwye, 2016*).

Our phylogenetic analysis contradicts the status of Tranatocetidae as a separate family by recovering *Tranatocetus* deeply nested within Cetotheriidae, as sister to *Metopocetus* (Fig. 11). The same applies to other presumed tranatocetids, including '*Aulocetus*' *latus*, '*Cetotherium*' *megalophysum*, and '*Metopocetus*' *vandelli* (*Gol'din & Steeman, 2015*). The monophyly of Cetotheriidae is primarily supported by the presence of a posteroventral flange on the compound posterior process (char. 184), and the parabolic outline of the postglenoid process (char. 118, in posterior view). Additional characters shared by all cetotheriids except *Tiucetus* and *Joumocetus* include ascending processes of the maxillae

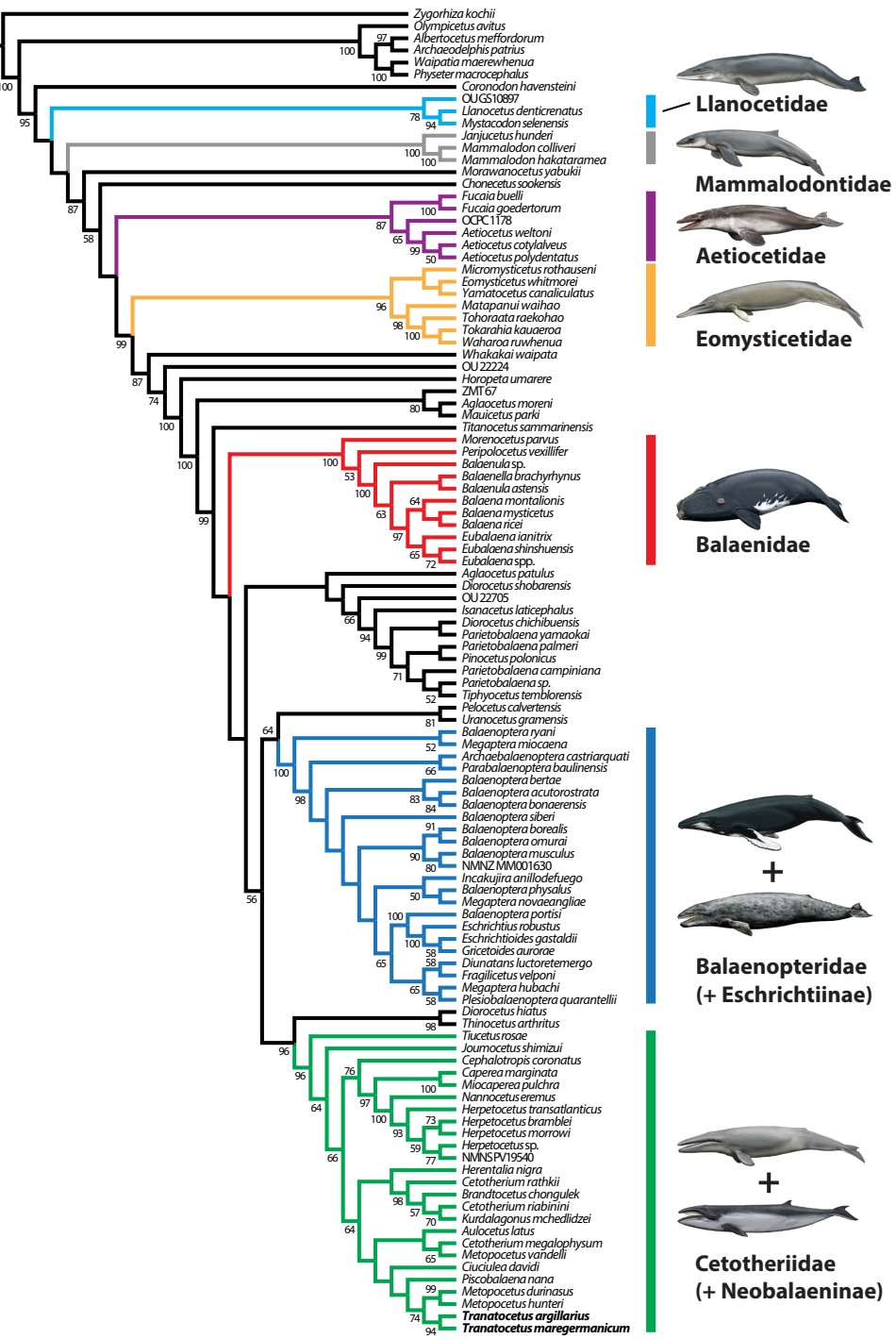

**Figure 11** Results of the total evidence phylogenetic analysis, showing the nesting of tranatocetids, including *Tranatocetus* itself, inside Cetotheriidae. Drawings of cetaceans by Carl Buell.

that directly contact the nasals (char. 67, reversed in cetotheriines), parietals that extend no further forward than the level of the postorbital process (char. 81), a distal surface of the compound posterior process that is firmly integrated into the lateral skull wall (char. 188), a squared anterior border of the tympanic bulla (char. 191), deep transverse creases on the dorsal surface of the involucrum (char. 207), and a mandibular body that increases in height towards the symphyseal area (char. 222, in lateral view).

Several features were previously noted as distinguishing tranatocetids from cetotheriids, including a smaller distal exposure of the compound posterior process, a narrower anterior portion of the tympanic bulla, a small angular process of the mandible, a low mandibular condyle, and a shallow glenoid fossa (*Gol'din & Steeman, 2015*).

The exposure of the compound posterior process is variable among cetotheriids, with the process being broadly exposed in herpetocetines, cetotheriines (sensu *Gol'din & Startsev, 2017*), *Herentalia*, *Metopocetus* and *Piscobalaena*, but less so in *Joumocetus* and *Tiucetus*. This range presumably is the result of a trend, with the basalmost taxa also having the smallest exposures (*Kimura & Hasegawa, 2010*; *Marx, Lambert & Muizon, 2017*). Tranatocetids fit different parts of this spectrum, with the exposed surface of *Tranatocetus* being comparable to that of *Herentalia* and *Metopocetus*, and far larger than in '*Aulocetus*' *latus*, '*Cetotherium*' *megalophysum* and '*Metopocetus*' *vandelli*. All tranatocetids furthermore share with cetotheriids a common structure of the compound posterior process, with the latter being expanded distally, and bearing a posteroventral flange which partially floors the facial sulcus (Figs. 7 and 12) (*Marx, Bosselaers & Louwye, 2016*; *Marx & Fordyce, 2016*). Tranatocetids therefore do not systematically differ in the morphology of their compound posterior process, but form part of morphological continuum encompassing all of Cetotheriidae.

Like the compound posterior process, the shape of the tympanic bulla is variable among cetotheriids. In ventral view, the anterior portion of the bulla is equal to or wider than the posterior half in *Herpetocetus* and cetotheriines, slightly narrower in *Ciuciulea*, and notably narrower in *Metopocetus* and *Piscobalaena* (Fig. 12). In most cetotheriids—in particular, cetotheriines and *Piscobalaena*—the anterior border of the tympanic bulla is furthermore notably squared. Tranatocetids generally conform to the narrow morphotype, including the squared anterior border, and thus tend to resemble *Piscobalaena*. (Fig. 12). We agree that narrowing of the anterior bulla may be a derived state setting apart certain species in the broader context of Cetotheriidae. Nevertheless, as shown by the striking resemblance of '*C.*' *megalophysum* and *Piscobalaena* (Fig. 12), there is no clear division between this morphology and that of several undoubted cetotheriids.

The angular process of the mandible tends to be enlarged in Cetotheriidae, either dorsoventrally as in cetotheriines (*Gol'din, Startsev & Krakhmalnaya, 2014*; *Gol'din, 2018*), or anteroposteriorly as in *Piscobalaena* and herpetocetines (*Bouetel & de Muizon, 2006*; *El Adli, Deméré & Boessenecker, 2014*). In the latter two, the process notably projects beyond the level of the mandibular condyle, and bears a well-developed fossa for the medial pterygoid muscle. Our new observations show that *Tranatocetus* precisely fits this elongate morphotype (Fig. 8), nesting it deep within Cetotheriidae. The mandibular

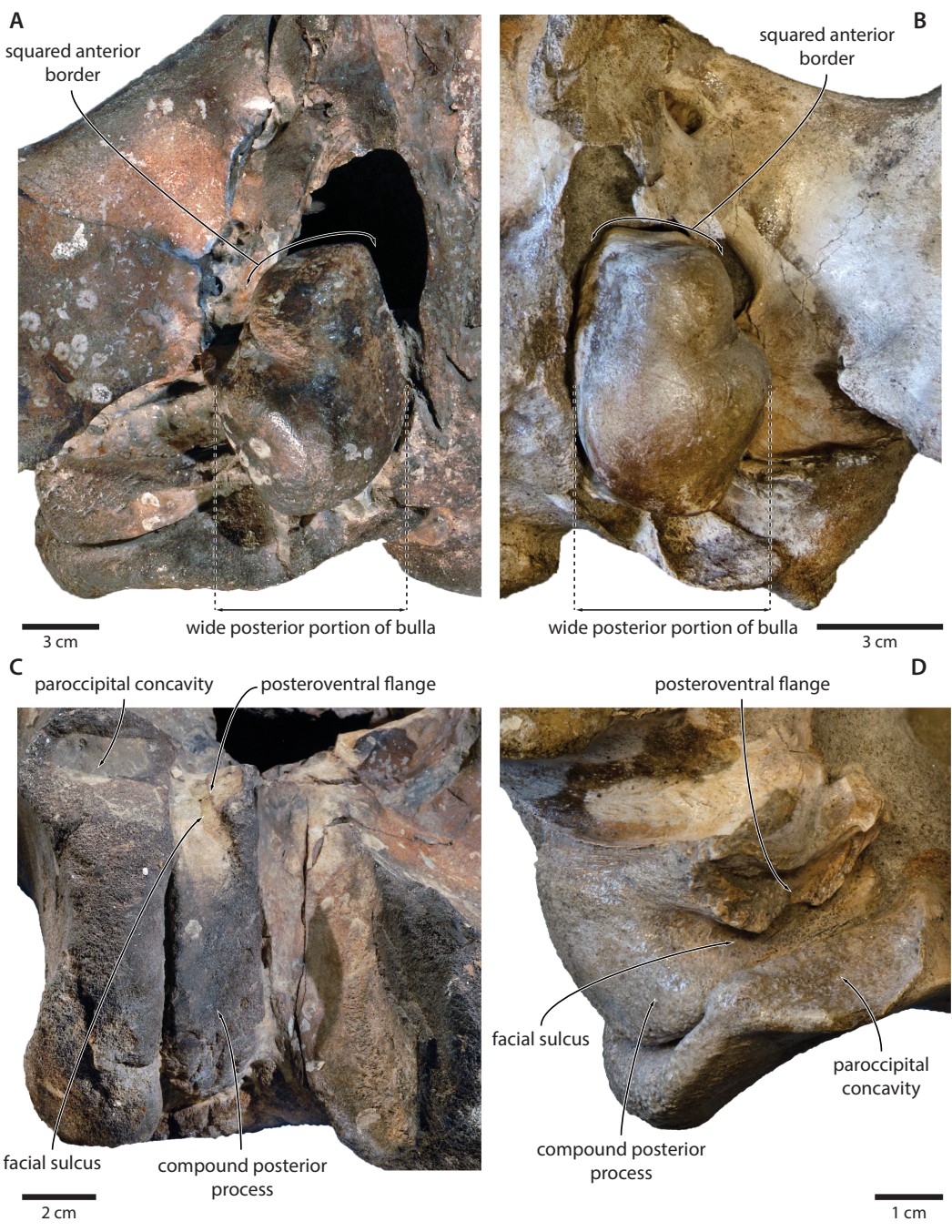

**Figure 12** Comparison of the auditory regions of (A, C) the presumed tranatocetid '*Cetotherium*' *megalophysum* (USNM 10593, holotype) and (B, D) the cetotheriid *Piscobalaena nana* (MNHN SAS 1616). (A) Right and (B) left auditory region in ventral view; (C) left compound posterior process in ventrolateral view; (D) right compound posterior process in oblique posterolateral view. Photographs by Felix G. Marx.

morphology of other 'tranatocetids' is poorly known, with no lower jaws having been described for '*A.*' *latus*, '*C.*' *megalophysum* and '*M.*' *vandelli*.

A comparatively small angular process, as previously suggested for tranatocetids (*Gol'din & Steeman, 2015*), appears to be the plesiomorphic state, based on its occurrence in both balaenopterids and a variety of Miocene non-cetotheriids (*Kellogg, 1934*; *Kellogg, 1968b*; *Steeman, 2009*; *Tanaka, Ando & Sawamura, 2018a*). In light of this observation, we question its usefulness as a distinguishing characteristic of tranatocetids, and predict that basal cetotheriids, such as *Joumocetus* and *Tiucetus*, may ultimately be revealed to have a markedly smaller angular process than other members of the family.

Like a small angular process, a low mandibular condyle appears to be a plesiomorphic feature characterising both balaenopterids and a broad range of Miocene non-cetotheriid mysticetes (*Kellogg, 1934*; *Kellogg, 1968a*; *Kellogg, 1968b*; *Kimura, 2002*; *Tanaka, Ando & Sawamura, 2018a*). By contrast, the condyle of cetotheriids tends to be somewhat elevated above the mandibular neck. The position and orientation of the condyle in turn correlates with that of the glenoid fossa of the squamosal, with the latter reportedly being relatively shallow in cetotheriids (*Gol'din, Startsev & Krakhmalnaya, 2014*; *Gol'din & Steeman, 2015*).

*Tranatocetus* has a posteriorly oriented, non-elevated condyle, and—in this regard— shows a relatively primitive morphology of the craniomandibular joint (Figs. 8 and 9). This anatomy could plausibly be plesiomorphic but, considering the otherwise clearly cetotheriid shape of the ramus, might also represent a secondary feature, perhaps associated with large body size. The anatomy of '*A.*' *latus*, '*C.*' *megalophysum* and '*M.*' *vandelli* remains unknown. As with a small angular process, we predict that a low mandibular condyle also primitively occurred in some basal cetotheriids, and hence is of limited value in distinguishing the latter from tranatocetids.

Overall, the evidence supporting a separation of Cetotheriidae and Tranatocetidae is thus relatively weak. By contrast, our new observations on *Tranatocetus* reveal a marked resemblance of this genus with several undoubted cetotheriids, borne out by the results of our phylogenetic analysis. These results cast doubt on the status of Tranatocetidae as a separate clade, and instead imply the existence of a single, extended family Cetotheriidae, including *Tiucetus* as its basalmost form.

## Implications for cetotheriid palaeobiology

At an estimated body length of 7.7 m (based on *Lambert et al., 2010*) to 8.7 m (based on *Pyenson & Sponberg, 2011*), *Tranatocetus maregermanicum* is the largest formally described cetotheriid. In stark contrast, most of the remaining members of the family do not exceed 5 m in length, and thus are relatively small compared to other mysticetes (*Bouetel & de Muizon, 2006*; *El Adli, Deméré & Boessenecker, 2014*; *Gol'din & Startsev, 2017*; *Gol'din, 2018*; *Slater, Goldbogen & Pyenson, 2017*). There are, however, notable exceptions, including *Herentalia nigra*, '*Cetotherium*' *megalophysum*, an as yet unnamed species from Peru, and a fragmentary skeleton from northern Belgium (*Bisconti, 2015*; *Bosselaers et al., 2004*; *Collareta et al., 2015*; *Slater, Goldbogen & Pyenson, 2017*).

In general, larger cetotheriids appear to cluster in the early Late Miocene, whereas smaller forms –in particular, herpetocetines, and *Piscobalaena*—dominate during the latest

Miocene and Pliocene (*Bouetel & de Muizon, 2006*; *El Adli, Deméré & Boessenecker, 2014*; *Tanaka, Furusawa & Barnes, 2018b*; *Tanaka & Watanabe, 2018*; *Whitmore Jr & Barnes, 2008*). Larger size plausibly correlated with a different ecological niche, with *Tranatocetus* perhaps being more pelagic than other cetotheriids, or targeting free-swimming schooling fish instead of benthos. The same may have applied to other large cetotheriids (*Collareta et al., 2015*), and possibly suggests a Late Miocene shift towards a different diet, habitat or feeding strategy.

The reasons behind this shift, if indeed it occurred, remain obscure, but it seems noteworthy that it coincided with the initial diversification of rorquals. In the Pisco Formation of Peru, for example, rorquals became locally abundant, and represented by two to three different species (*Di Celma et al., 2017*), at the same time as cetotheriids declined in number and size (*Bianucci et al., 2016a*; *Bianucci et al., 2016b*). We suggest that this phenomenon might be explained by niche partitioning between small, suction feeding and possibly benthic/neritic cetotheriids on the one hand (*El Adli, Deméré & Boessenecker, 2014*; *Gol'din, Startsev & Krakhmalnaya, 2014*), and large, lunge-feeding, pelagic rorquals on the other. Cetotheriids continued to occupy the 'small (benthic) filter feeder' niche for the remainder of the Miocene and Early Pliocene, but then largely disappeared-alongside small balaenids and most small rorquals- with the onset of Northern Hemisphere glaciation around 3 Ma (*Marx & Fordyce, 2015*; *Slater, Goldbogen & Pyenson, 2017*).

In the Northern Hemisphere, grey whales currently occupy a niche similar to that inferred for cetotheriids, which may help to explain their convergent skull morphologies (*Bisconti, 2008*; *Steeman, 2007*). The oldest described eschrichtiine material dates to the Late Miocene, but is currently restricted to a fragmentary mandible (*Bisconti & Varola, 2006*). More confidently identified specimens are only known from the Pliocene (*Bisconti, 2008*; *Ichishima et al., 2006*; *Kimura, Hasegawa & Kohno, 2017*; *Whitmore Jr & Kaltenbach, 2008*), raising the question for how long, and how broadly, cetotheriids and eschrichtiines overlapped. Competition between the two could plausibly have limited the resources available to cetotheriids further, even though they seemingly were more common and had a wider distribution (including in the Southern Hemisphere) (*Bouetel & de Muizon, 2006*). Further insights into when and where eschrichtiines acquired their modern morphology may help to answer this question.

### Institutional abbreviations

**GMUC**   Geological Museum of the University of Copenhagen, Denmark
**MNHN**   Museum National d'Histoire Naturelle, Paris, France
**NMR**   Natuurhistorisch Museum Rotterdam, the Netherlands
**USNM**   United States National Museum of Natural History, Washington DC, USA

## ACKNOWLEDGEMENTS

We thank Remie Bakker and Tone Skelton for hosting us, and their help in mounting the specimen; Bram Langeveld and Henry van der Es for their assistance at NMR; Mette Steeman for the warm reception at Gram; Pavel Gol'din for helpful discussions

on cetotheriid systematics; Carl Buell for providing illustrative drawings of living and fossil cetaceans; Mette Steeman and Toshiyuki Kimura for their constructive reviews; and the staff of all of the institutions involved for access to material and help during our visits.

### Funding

Felix Georg Marx was supported by a Fonds de la Recherche Scientifique (FNRS) postdoctoral fellowship (32795797), and an EU Marie Skłodowska-Curie Global Postdoctoral fellowship (656010/ MYSTICETI). The funders had no role in study design, data collection and analysis, decision to publish, or preparation of the manuscript.

### Grant Disclosures

The following grant information was disclosed by the authors:
Fonds de la Recherche Scientifique (FNRS) postdoctoral fellowship: 32795797.
EU Marie Skłodowska-Curie Global Postdoctoral fellowship: 656010/ MYSTICETI.

### Competing Interests

The authors declare there are no competing interests.

### Author Contributions

- Felix G. Marx conceived and designed the experiments, performed the experiments, analyzed the data, prepared figures and/or tables, authored or reviewed drafts of the paper, approved the final draft.
- Klaas Post conceived and designed the experiments, performed the experiments, analyzed the data, contributed reagents/materials/analysis tools, authored or reviewed drafts of the paper, approved the final draft.
- Mark Bosselaers performed the experiments, analyzed the data, authored or reviewed drafts of the paper, approved the final draft.
- Dirk K. Munsterman performed the experiments, analyzed the data, contributed reagents/materials/analysis tools, authored or reviewed drafts of the paper, approved the final draft.

### Data Availability

The phylogenetic matrix and dinoflagellate cyst counts are available as part of the Supplementary Material.

### New Species Registration

The following information was supplied regarding the registration of a newly described species:
Publication LSID: urn:lsid:zoobank.org:pub:D39ACC32-687F-4C95-9CD8-A2B17B2DBAFC;
*Tranatocetus maregermanicum* LSID: urn:lsid:zoobank.org:act:499F1C5C-3C3F-48A9-AD97-AF19F99DE886.

## Supplemental Information

Supplemental information for this article can be found online at http://dx.doi.org/10.7717/peerj.6426#supplemental-information.

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
