# Peer review of "A large Late Miocene cetotheriid (Cetacea, Mysticeti) from the Netherlands clarifies the status of Tranatocetidae"

_PeerJ, doi:10.7717/peerj.6426_

## Round 0.1 · original submission · Minor Revisions

Dear authors,

I have accepted the decision of ‘minor revisions’ from the reviewers. Please address their comments in detail.

·

Basic reporting

no comment

Experimental design

no comment

Validity of the findings

no comment

Additional comments

The manuscript by Marx et al. " A large Late Miocene cetotheriid (Cetacea, Mysticeti) from the Netherlands clarifies the status of Tranatocetidae" represents interesting additions to our knowledge of cetacean evolution. The authors described a new species of Cetotheriidae and they tested the idea that the Cetotheriidae and enigmatic taxon Tranatocetidae indeed form separate clades. Based on the several morphological characters indicated by authors and the result of phylogenetic analysis, the authors clearly indicated that Tranatocetidae and Cetotheiirade should be integrated into single family, Cetotheriidae.

The description of the specimen is detailed and the discussion is clear and solid. I think this paper will be widely received by people interested in the evolution of marine mammals. I have a few comments and suggestions that I think it will improve the paper, all of them are minor.


Lines 37-44: three distinct morphotypes of Cetotheriidae
The authors mentioned that at least three distinct morphotype is found in the Cetotheriidae. However, the authors just mentioned on body size and fossil occurrence of each morphotypes and not mentioned on the morphological characteristics for each morphotypes. Please indicate the morphological characteristics for each morphotypes.

Line 124
The words "the level of the parietal" should be replaced by "the level of the anterior edge of the parietal"?

Lines 130-131: sharp cranial rim
The authors described as "a sharp cranial rim surrounding the proximal opening of the facial canal", but it is not shown in the figure. Please indicate it in the figures.

Lines 173-180: diagnosis
This part of diagnosis is on the diagnostic characters for genus Tranatocetus, so, perhaps, this part should be included in the section of the emended diagnosis of the genus.

Line 180: diagnosis
The authors mentioned here that "Differs from T. argillarius in being slightly larger". I think this might include ambiguity; it is difficult to clearly suggest that the slight body size difference is not due to the individual variation, but due to the phylogenetic distinction.

Line 271
The words "paroccipital cavity" should be replaced by "paroccipital concavity"?

Lines 493-499: niche partitioning, extinction of the Cetotheriidae
The eschrichtiids are also known as benthic filter feeder, and several early members of the Eschrichtiidae have been recovered from the Pliocene deposits. Perhaps, they might have occupied similar ecological niche (small, benthic suction feeder). Do they have also an impact on the extinction of the Cetotheriidae?


Toshiyuki Kimura
Gunma Museum of Natural History

·

Basic reporting

No comment

Experimental design

No comments

Validity of the findings

No comment

Additional comments

I find the text well written and well structured, concise and to the point. Particularly, I find the description well written and the figures are supporting the text very nicely. The authors argue well for their opinions on both the taxonomic position of the new material and the overall phylogeny of the groups investigated and have used standardly accepted methods to reach their conclusions. I find only minor suggestions, mainly to improve readability.

---

## Round 0.2 · accepted · Accept

Dear authors,

I am pleased to tell you that your manuscript has been accepted for publication.

Production staff will shortly be in touch with you regarding your proofs.

Once again, thank you for choosing PeerJ as your publishing venue, and I hope you use us in the future.

# ·

Basic reporting

no comment

Experimental design

no comment

Validity of the findings

no comment

Additional comments

The authors carefully and appropriately revised their manuscript and implemented all of my previous suggestions. On the author's response to my previous comment on the diagnosis, I agreed the author's response. I do not find any problem, and now I think this manuscript is ready for publication.

Toshiyuki Kimura
Gunma Museum of Natural History